# Interactions between Stress Levels and Hormonal Responses Related to Sports Performance in Pro Women’s Basketball Team

**DOI:** 10.3390/jfmk9030133

**Published:** 2024-07-31

**Authors:** Álvaro Miguel-Ortega, Julio Calleja-González, Juan Mielgo-Ayuso

**Affiliations:** 1Faculty of Education, Alfonso X “The Wise” University (UAX), 28691 Madrid, Spain; 2International Doctoral School, University of Murcia (UM), 30003 Murcia, Spain; 3Physical Education and Sport Department, Faculty of Education and Sport, University of the Basque Country (UPV/EHU), 01007 Vitoria-Gasteiz, Spain; julio.calleja.gonzalez@gmail.com; 4Faculty of Kinesiology, University of Zagreb, 10110 Zagreb, Croatia; 5Faculty of Health Sciences, University of Burgos (UBU), 09001 Burgos, Spain; jfmielgo@ubu.es

**Keywords:** basketball, hormones, performance, female

## Abstract

The testosterone to cortisol ratio (T:C ratio) is a measure of whether elite athletes are recovering from their training. This study described this hormone balance stress in elite women’s basketball. (1) Objectives: to analyse the fluctuation of T:C ratio over a 16-week period and explore itis relation to their athletic performance. The participants characteristics were: (height: 177.6 ± 6.4 cm; body mass: 77.808 ± 12.396 kg age: 26.0 ± 5.9 years; and a playing experience of 14.7 ± 2.9 years with 5.0 ± 1.2 years at the elite level. The T:C ratio at Time 1 is: 4.0 ± 2.4 (*n* = 12); and at Time 2 is: 5.1 ± 4.3 (*n* = 12). (2) Methods: during 16 weeks of competition, participants underwent analysis of blood samples to assess various biochemical parameters including hormone levels. In addition, their athletic performance was assessed with the following tests: jumping (SJ, CMJ, ABK, DJ); throwing test with a medicine ball (3 kg); Illinois COD agility test; sprint repeatability with change of direction; 20-m speed test without change of direction; and Yo-yo intermittent endurance test IET (II). (3) Results: The main alterations observed were an increase in T levels (1.687%) and a decrease in C levels (−7.634%) between moments, with an improvement (26.366%) in the T:C ratio. Improvements were also observed in some of the tests developed, such as jumping (SJ: 11.5%, *p* = 0.029; CMJ: 10.5%, *p* = 0.03; DJ: 13.0%, *p* = 0.01), upper body strength (MBT: 5.4%, *p* = 0.03), translation ability (20 m: −1.7%), repeated sprint ability (RSA: −2.2%), as well as intermittent endurance test (Yy (IET): 63.5%, *p* = 0.01), with significant changes in some of the performance tests. (4) Conclusions: T:C ratio may differ in a manner unrelated to training volume, showing some variation. These results may be attributed to the accumulation of psychophysiological stress during the season.

## 1. Introduction

One of the main aims of high-level sport is to prescribe adequate training loads to maximise athletes’ performance during the competition period, avoiding excessive overload and/or inadequate recovery that could affect athletes’ health. In order to maintain athletes wellbeing status, variables including the participant’s training state, exercise mode, intensity, and duration must be considered [1].

Conversely, exercise has been shown to be a potent endocrine system stimulant, as evidenced by the hypothalamic-pituitary-adrenal axis [2]. In addition, post-exercise modification in hormones concentrations can positively influence the musculoskeletal system and thus improve the performance of athletes [3].

In this sense, exercise intensity and duration have a major impact on the rates at which muscle glycogen—a type of carbohydrate that gradually decreases over time because its concentration in skeletal muscle is finite—degrades [4]. Dependence on specific energy substrates increases exponentially with exercise intensity. This aspect also largely conditions the hormonal response [5].

Studies on the endocrine system and its response to exercise and sport have increased considerably in the last 25 years [6], as well as interest in using exercise as a therapeutic approach to reduce the effects of ageing and enhance health [7]. A younger endocrine profile may be promoted as one of the fundamental processes by which training attenuates numerous indices of biological ageing [8]. However, the physiological effects of exercise and physical activity on hormones that somehow affect the processes that regulate glycogen levels in adult athletes remain comparatively understudied despite this growing interest, and randomised trials of high methodological rigour and quality are scarce or non-existent for some hormones. According to several studies, the hormone profile changes significantly depending on the type and extent of exercise [9]. Some authors suggest individualised monitoring of the hormone profile [10] to optimise training prescription and even assess the training potential of female athletes [11,12].

In a particular, female athletes have an additional hormonal profile, which is the menstrual cycle, an individual variable that causes the state of each athlete to change throughout the cycle depending on her hormonal profile and which must be considered when planning training and quantifying the load if the aim is to obtain maximum performance from each athlete [13].

Therefore, a poorly planned load negatively affects the technical and tactical performance of athletes, increasing mental fatigue and stress, aspects that modulate the endocrine response [14]. Even more so if this planning is done longitudinally and if the participants are women, where the number of studies and results is quantitatively lower than for men. To effectively manage and potentially adjust this allocated load as needed throughout the annual plan, it is essential to accurately quantify players’ workloads [15].

Concretely, in basketball is a fast-paced, high-contact sport that presents on a variety of key performance parameters related to strength, speed, agility, endurance, specialised skills or stress that can also affect a player’s ability to heal from injuries and improve their technical skills. In addition, the competition schedule often includes competitions two or three times a week [16,17], which requires a thorough understanding of the effects of training and competition to establish an appropriate training periodization [18]. Although basketball is a team sport, it is important to consider the principle of individualisation [19] to develop the specific functions associated with each position and playing time in competition Given that every athlete is unique, it is critical to manage the internal stress that comes with training and competition [20].

In basketball it is vital to closely monitor player effort levels during the pre-season and regular season [21]. This practice allows coaches to strategically allocate and schedule workloads, ultimately promoting beneficial physical and physiological changes [15].

Therefore, these elements are necessary to choose the best course of action, reduce any potential negative consequences of training and competition, modify training to guarantee proper recovery mechanisms, prevent fatigue build up, or make up for lack of stimulation [22].

In line with these premises, the mechanisms associated with fatigue produced by a decrease in protein catabolism and a halt in lipolysis, which prevent the mobilisation of energy reserves [23]. So, in basketball, fatigue-related stress can have an impact on shooting ability and muscle recovery [24]. Both factors, stress and fatigue, cause an increase in cortisol (C) levels and a decrease in serum testosterone (T) levels [25].

Thus, T, which serves as an indicator of the body’s regeneration rate [26], C, whose blood levels are thought to be a good indicator of how much accumulated stress there is, and the T:C ratio, which reflects the anabolic/catabolic balance [27] and which tends to increase with fatigue, are suggested as indicators of how well anabolic and catabolic processes are balanced [28] to identify overtraining and/or stop competitively-induced excessive psychophysical activity [29]. 

Regardless of the type of exercise, the anabolic ratio may be more helpful for tracking an athlete throughout the season as a measure of performance improvement. This is because metabolic disturbances related to energy generation and utilisation [27] reduced skeletal muscle performance and may be caused by factors including iron insufficiency and catabolic/anabolic imbalance.

Crucially, it has also been noted that frequent exercise without enough rest time might cause this ratio to persistently change [30]. Low levels of T and high levels of C have been linked to overtraining and decreased performance, whereas high levels of T have been linked to optimal performance and recovery processes [25]. On the other hand, it has been observed that when an overall imbalance exists over a prolonged period, it can lead to fatigue, overexertion, and overtraining syndrome. This is because proper balance between training load (intensity and volume) and recuperation is necessary for physical exercise to increase performance. 

In recent years, some basketball teams strategically rest their players during games as load control and recovery have become important issues [31]. Various physical tests have become fundamental tools for monitoring players’ physical condition. For this reason, the tests performed—jumping [32], medicine ball throw, Illinois agility test [33], sprint repetition skill with change of direction [34], 20 m run without change of direction [35] and the Yo-yo intermittent endurance test [36]—have a direct relationship with blood T and C levels [37]. Since these hormonal parameters are key to understanding the body’s response to exercise and the stress that players are subjected to during matches, continuous monitoring is essential to adapt training programs and ensure the right balance between load and recovery [38].

From this point of view, deepening the knowledge of the impacts observed in the biomarkers throughout a competitive season may offer possibilities for improving intervention strategies at the individual level [39] and, therefore, at the collective level. However, despite this and as mentioned above, only a small number of publications have addressed hormonal responses in women’s basketball [40], for the best of the authors knowledge.

To develop unique training and work dynamics for professional basketball female players, the main aim of this descriptive study was to investigate the fluctuations of plasma total T and C levels and their ratio over the course of a regular competitive season (16 weeks) of elite female basketball players.

## 2. Materials and Methods

### 2.1. Participants

Twelve female basketball players (Table 1) were assessed in the 2018–2019 season. These athletes competed in the “Liga Dia”, which was overseen by the Spanish Basketball Federation (FEB), at the greatest level.

The *n* is a pure opinion sample, chosen at the discretion of the research team on a non-probabilistic and non-random basis to indicate whether a particular trait or characteristic exists in each population [41] at an exploratory level to obtain a more accurate estimate. Thus, the participants represented different nationalities. The team provided players from several national teams: 3 players from the Spanish national team, 1 from the Croatian national team, 1 from the Senegalese national team, and 1 from the Swedish national team. With contests excluded, a working weeks’ worth of training was, on average, 22.5 h. This comprised three double training sessions, consisting of 180 min of physical exercise on Monday, Wednesday, and Thursday, plus 150 min of technical–tactical work in the morning. Furthermore, in addition to physical components, there were two 180-min sessions on Tuesday and Friday that concentrated on things that were solely tactical or technical–tactical. On Saturday or Sunday, there were match days, and then there was a rest day.

To be part of the study, participants had to meet certain requirements: a) practice elite sports for at least one year; b) not have had serious injuries in the last 30 days that prevented them from training; and c) attend all data collection sessions.

No one who took part in the data gathering has ever had allergies, hormone abnormalities, or injuries. They were also forbidden from using prescription pharmaceuticals or illegal substances that would alter their body mass. Before the examination began, they were instructed to refrain from physically demanding activities for at least a day. The parents of the minor participants gathered and signed consent papers. In addition to the prescribed training intervention, the participants were directed to abstain from any physically demanding activities throughout the trial. Throughout the trial, the participants were also urged to refrain from drug use and to continue with their regular sleep, food, and water routines.

### 2.2. Exclusion Criteria

Female players were properly told about the experimental methods, potential hazards, and advantages before the trial. In accordance with the ethical standards outlined in the 2013 World Medical Association Declaration of Helsinki [42] for the conduct of medical research involving human subjects, they were requested to provide written consent before participation. The project (number: M10_2017_216) was authorised by the University of the Basque Country’s Human Research Ethics Committee. The study complied with the guidelines set forth in the Organic Law 15/1999 on 13 December 1999, which concerned the Protection of Personal Data.

### 2.3. Experimental Design of the Problem

To ensure ecological validity, the present study was conducted in non-experimental settings [43]. This is a descriptive study developed to collect information to demonstrate the relationships between the described hormones and performance in female team sports. Thus, neither the coaching staff nor the participants received any direction or intervention from the research team. The team only received training data, competition schedules, and match outcomes from the coaching staff.

### 2.4. Evaluation Plan

Assessments were conducted at two points in the season (T1: September 2018, week 1 of mesocycle 1 of macrocycle 1, pre-season; T2: January 2019, week 13, mesocycle 1 of macrocycle 2, first competitive break) just prior to a training session. These times were selected given that the measured variables were expected to exhibit considerable variability at these points [44].

To rule out any biological or environmental influences that could have an impact on the outcomes, measurements were taken at the sports facilities where the players trained and competed. The measurements were taken in San Sebastián, Gipuzkoa, Spain’s Municipal Sports Centre “*José Antonio Gasca*”. The first session had a humidity of 71% and a temperature of 28.7 °C, while the second session had a humidity of 47% and a temperature of 16.4 °C. This represented a significant difference of 42.86% in humidity and 12.3 °C in temperature between these two times. These differences are due to the seasonal changes at the different time points of data collection, although these differences may influence the physiological level of muscle viscosity, which may affect the variability of warm-up intensity [45].

To obtain accurate results, participants were given specific instructions to follow prior to analysis. The day before the test, they were told not to do any physical exercise at all. In addition, they were advised not to eat or drink anything for 4 h before the test and to make sure they were properly hydrated. In addition, participants were asked to go to the toilet 30 min before the data collection process began (Figure 1).

#### 2.4.1. Blood Test [46]

Prior to each physical testing session, a blood test was performed at both time Prior to each physical testing session, a blood test was performed at both data collection periods. A university registered nurse was present to collect, label and package the blood samples for biochemical analysis. The Vacutainer vacuum system^®^ with anticoagulant was used to collect the blood samples. The tubes were hermetically sealed prior to transport to ensure that the samples were safely transported to the laboratory. This precaution prevented any jolts or sudden movements that could cause haemolysis of the blood. Whole blood samples with anticoagulant were transported refrigerated using cold accumulators, which maintained the temperature between +4 and +8 °C. Samples arrived at the laboratory within 24 h from the time of collection. Analyses were carried out at a specialized center in clinical analysis. The laboratory where the analyses were performed specializes in examining the fluids of the human body to aid in the prevention, diagnosis and treatment of disease. It is staffed by a diverse team of professionals and meets quality standards including medical analysts, nursing graduates, laboratory technicians and administrative staff. In addition, it follows a quality management system in accordance with ISO 9001:2008, which was also used in this long study [47].

#### 2.4.2. Kineantropometry

Height measurement was determined using the Holtain^®^ stadiometer (Holtain Ltd., Dyfed, UK) with millimetre accuracy. The measuring range is from 60 to 209 cm. While weight was obtained using an electronic scale with a SECA^®^ scale (Seca Corp., Hanover, MD, USA) (accuracy: 0.1 kg; range: 2–130 kg). BMI was calculated using the formula BM/height^2^ (kg/m^2^). Body fat percentage was assessed by skinfold measurement taken in triplicate using a Harpenden^®^ skinfold caliper and analyzed by two observers. The sum of 8 skinfold measurements (mm) (biceps, triceps, subscapular, iliac crest, supraspinal, abdominal, anterior thigh and calf) was calculated and the equation developed by Carter & Heath [48] was used to determine the body fat percentage of the different somatotypes (mesomorphy, ectomorphy and endomorphy) in women % fat mass = 3.5803 + (Σ8 skinfolds × 0.1548).

#### 2.4.3. Physical Performance Test

These tests have been chosen given that they provide objective and accurate information on the skills explored, and according to the literature, they are valid, reliable, objective, and standardisable, and are commonly used in basketball.

Jumping (SJ, CMJ, ABK, DJ)

A Chronojump^®^ Bosco System DIN-A1 device (Chronojump Association for the Research and Dissemination of Technology Applied to Physical Activity and Sport, Barcelona, Spain) [49], which was controlled by a time measuring device, was used to record the fitness test data. The microcontroller had an error of 0.1%, which ensured a validity of 0.95 (CCI). The measurement device was validated by the ACSM [50] with accuracy over the entire range investigated, presenting an average error of 0.04 ± 0.18% for low signals (such as contact time in a jump) and 0.05 ± 0.19% for high signals (flight time) [51].

Squat jump (SJ).

During this exercise, the maximal concentric dynamic strength of the lower limbs is assessed. Its similarities to basketball are related to its ability to jump and accelerate. It serves as a gauge for the proportion of fast fibres. This manifestation is supplemented by a second “contractile capacity” component that relates to the ability to synchronise the contraction of the fibres for a more consistent value of instant recruitment.

This examination shows consistent results across different trials, with a small amount of variability (3.0%). It also demonstrates strong reliability scores (0.97 and 0.98). The test displays a consistent relationship between its results over time (0.91), as well as a high degree of correlation within the same subjects (0.97). The range of individual variations in this assessment is between 2.4% and 4.6% [52].

Its ratio is the highest slow force produced during a Hill’s Law-based slow concentric exercise (e.g., 45° or squats). When performing the leap, considerable care must be made to avoid repulsion before the jump and to begin from a static half-squat position to decrease the margin of error.

Countermovement jump (CMJ) [53]

Participants were instructed to leap as high as they could while making a fast countermovement, maintaining their hands on their hips, starting from a standing posture with both feet together. The flight time was used to compute the change in the height of the body’s centre of gravity. The take-off and landing positions of the body’s centre of gravity were considered while determining the jump height. Every competitor was allowed two attempts, separated by a one-minute rest time. The finest trial was noted. Very little inter-trial variability (coefficient of variance of 3.0%) is a defining feature of the CMJ. The greatest correlation (r = 0.87) and best factorial validity are exhibited by the CMJ test with respect to the explosive power component. Based on these results, it can be said that one of the most reliable field tests for estimating lower limb explosive strength is the CMJ [52].


Abalakov jump


To determine the “reflex-elastic-explosive” manifestation, this test measures the lower limb’s maximal power as well as its explosive strength. This period, not the acceleration phase is when the stretch reflex is released because of the duration of the exercise and the fact that around 50% of it is spent damping (mainly eccentric). The two heights that the arms generate may be measured, and the percentage difference between the heights attained in the Abalakov and the CMJ is what we refer to as the arm utilisation ratio. A component analysis of all the jump tests shows that this one accounts for 82.90–95.79% of the variance. The jump test has a low coefficient of variation (1.54–4.82%) and a high correlation coefficient (0.969–0.995) [53].

Drop jump [54]

This is a leap that is executed with the legs extended and going downhill after falling from a specific height. The hands must be on the hips and the trunk straight during the continuous motion to verify and evaluate the “re-flex-elastic-explosive” manifestation of strength. Reliability is attained with a coefficient of variation of 3.6–6.4%, an intra-class ratio coefficient of 0.70–0.92, and a measurement standard error of 8.5–18.4 microseconds (ms).

b.Throwing test with a medicine ball (MBT; 3 kg)

The conditional ability of strength is a key factor in basketball when analysing the aspects of the effort in this sport modality, which consists of quick yet intense efforts that lead to athletic performance [55]. The two tests most used in this sport to track and assess the effects of training on the upper body are usually the medicine ball throw for power evaluation or the bench press 1RM strength test [56].

The Test–retest reliability for the medicine ball throwing test, which assessed upper body strength, was r = 0.98 even though r = 0.96 had previously been found. The means of each throwing modality (standing, kneeling, sitting, and one-handed) showed an intra-class correlation coefficient (ICC) of 0.98 and a correlation value of r = 0.49, *p* ˂ 0.01, respectively [47], thus demonstrating test–retest validity [57].

c.Illinois COD agility test, sprint repeatability with change of direction, and speed test without change of direction (20 m).

Photoelectric cells from the MicroGate^®^ Witty Wireless Training Timer were used in these investigations. This device features a minimum resolution of 0.125 ms, redundant coding, an event latency of 1 ms, and an accuracy of 0.4 ms for pulse transmission.

20-m speed test without a direction change.

This test has been used as it is the standard goal test in terms of reaction speed coupled with the ability to express the maximum overall cyclic translation speed. Starting from a standing still posture, participants could undertake two attempts of the 20 m sprint, with the quickest attempt being recorded. The start and finish lines of the sprint were marked using Witty-Gate photocells from MicroGate^®^ Timing Systems S.R.L. in Bolzano, Italy. The sprint was raced around the border of the basketball court. Test–retest correlation coefficient of 0.91 indicated high levels of reliability (no need for pre-testing) for the 20-m sprint test, which had intra-class correlation indices of 0.11–0.49 and coefficients of variation of 16.8–51.0% [58].


Repeated sprint ability test with change of direction (RSA)


Repetitive sprinting ability (RSA) has long been considered a type of movement indicative of high-intensity motions performed by athletes participating in team sports. Though it has been disputed, the importance of RSA as a crucial physical aspect of performance in intermittent sports is postulated. Most intermittent sports rely heavily on technical and tactical proficiency to achieve their goals. Furthermore, the ability to repeat sprints has been connected to the emergence of fatigue in various activities; the total amount of time spent repeating a sprint varies by 2.3% between persons. Participants performed 7 repetitions of running at maximum intensity (34.2 m), with active recovery pauses of 25 s between each repetition [59].


Illinois COD agility test


Due to its high validity and reproducibility [60], the Illinois Agility Test, recognised as a standard agility test, has been used to measure the agility and speed skills of basketball players in responding, accelerating, decelerating, and changing the direction of movement. The test has a standard error of measurement of 0.19 s and an intra-class correlation value of 0.96 (95% CI, 0.85–0.98). A *t*-test to determine the validity of the COD IAGT yields the following results: r = 0.31 [95% CI, 0.24–0.39]; and *p* < 0.05, suggesting that the test appears to be reliable and valid [61].

d.Yo-yo intermittent endurance test IET (II)

The endurance test was conducted using the Yo-Yo Pro-4.49 programme, which is intended for advanced group and individual testing (Recovery Level 1, 2, and Endurance Level 1, 2). The results of this test have also been used to indirectly calculate maximal oxygen uptake (VO_2max_) [62].

There is currently a strong trend in team sports to assess aerobic fitness using incremental and intermittent testing with pauses because of the 20m-SRT. One clear example of this is the use of the YOYO IET test at any of its two levels. Therefore, the literature suggests testing two things with this test: the capacity to recover from this type of exercise and/or the ability to repeat high-intensity intermittent efforts. Its applicability and validity have thus been looked at in several team sports, including basketball. The correlation between the total metres run throughout the test and the total metres run during high intensity runs (runs over 15 km/h) are what determine the validity of this test. The r-value for relationships pertaining to basketball is 0.77.

### 2.5. Statistical Analysis

Every data point is displayed as mean ± SD. The Shapiro-Wilk normality (<30) test was employed to determine if the variables were normal. For repeated or paired measurements, the parametric Student’s *t*-test was applied if the data had a normal distribution. To calculate the percentage change (%) of outcome variables from T1 to T2, the formula [(T2 T1)/T1] × 100 was utilised. With *n* = 12, the G*Power programme (version 3.1.9.7) was utilised to compute the sample size (SS = 11.86), statistical power analysis, and effect size. Partial eta squared (*η*^2^*_p_*), which measures the degree of influence between individuals, was calculated [63]. The following interpretation was used given that this metric is prone to overestimate the size of the impact: 0 ≤ *η*^2^*_p_* ≤ 0.05 indicates no impact, 0.05 ≤ *η*^2^*_p_* ≤ 0.26 indicates minor influence, 0.26 ≤ *η*^2^*_p_* ≤ 0.64 indicates moderate effect, and *η*^2^*_p_* ≥ 0.64 indicates considerable effect [63].

Probabilistic inference [64] has been used to analyse the data obtained and make predictions based on probabilities. By introducing this technique in the interpretation of information, the aim is to provide the reader with greater clarity and confidence in the results obtained in order to establish a better understanding of the possible interpretations and conclusions. In this way, critical and analytical reading is encouraged, in which not only the information presented is assessed, but also its level of reliability.

Furthermore, the Pear-son correlation analysis and the Hopkins [65] correlation magnitude were carried out. Trivial (r ˂ 0.1), tiny (0.1 < r ˂ 0.3), moderate (0.3 ˂ r ˂ 0.5), high (0.5 ˂ r ˂ 0.7), extremely high (r ˂ 0.9), near perfect (r > 0.9), and perfect (r = 1) were the classifications for correlation coefficient sizes. Subsequently, we propose the effect size variation, which offers a more comprehensible and practical approach to employing probabilistic reasoning to derive conclusions as well as a simple (albeit incomplete) depiction of the necessary change in sample size [66] based solely on the uncertainty surrounding the true value of the statistic. It has been widely acknowledged that the significance of the difference gained is substantial when it deviates from expectations in a way that cannot be believed to be created by chance alone, as the result is unlikely to be the product of chance or random fluctuation of this sort. As a result, only findings with a correlation coefficient higher than 0.5 will be considered. In accordance with accepted guidelines, the prevalence of being overweight in the study sample was determined using BMI. Measurement results were entered into an Excel spread sheet and then subjected to Windows SPSS^®^ 25.0 software (Inc., Chicago, IL, USA) for statistical analysis. A significance level of *p* 0.05 was used.

## 3. Results

### 3.1. Hormonespon Como en Resulatdos Blood Tests

Table 2 describes the percentage and absolute changes in all hormonal measurements and the corresponding ratio between the first test (T1) and the second test (T2). The primary alterations noted were a rise in T levels (1.687%) and a fall in C levels (−7.634%). As seen in Table 2, these modifications suggest a 26.366% improvement in the T:C ratio. But, as can be shown, there are no statistically significant differences given that the *p*-values are more than 0.05.

### 3.2. Kineantropometry 

The results at the kinanthropometric measurements between T1 and T2, were that body mass decreased (-2.645 ± 3.2 kg), body fat also decreased (-2.672% ± 3.3%), lean mass increased (2.151% ± 3.9%) and the Σ8 skin folds decreased (-4.395 ± 13.5 mm), all of them being non-significant changes. Establishing a somatotype that represents a high correlation with the ability to repeat sprints (r = 0.7860; *p* = 0.0024).

### 3.3. Performance Tests

Table 3 shows the changes in percentage and absolute terms for each performance metric for the players between times. The results of 77.8% of the performance tests showed improvements. Specifically, improvements were observed in some of the jumps (SJ: 11.5%; CMJ: 10.5%; DJ: 13.0%), upper body strength (MBT: 5.4%), translation ability (20 m: −1.7%), repeated sprint ability (RSA: −2.2%), as well as in the intermittent endurance test (Yy (IET): 63.5%). 

On the other hand, there was a slight decrease in performance in the ABK jump: −2.1% and in agility (Illinois: 1.6%). From a statistical point of view, significant changes were observed in several of the tests performed; in three of the four types of jumps performed (SJ: 11.5%, *p* = 0.029; CMJ: 10.5%, *p* = 0.03; DJ: 13.0%, *p* = 0.01); in upper body strength (MBT: 5. 4%, *p* = 0.03) and cardiorespiratory capacity (*p* = 0.008) with the significant improvements produced in the intermittent endurance test and its indirect extrapolation to VO_2max_.

### 3.4. Relationship between Hormone Levels and Performance Changes

At time T1, moderate correlations of the SJ jump with C (r = 0.333; *p* = 0.291) and with T (r = −0.310; *p* = 0.326) were observed. Of the DJ with the T:C ratio (r = −0.420; *p* = 0.068). Of the medicine ball throw with the C (r = −0.445; *p* = 0.634), with the T (r = 0.307; *p* = 0.331) and with the T:C ratio (r = 0.409; *p* = 0.186). At the same time point, high correlations were obtained for DJ with C (r = 0.633; *p* = 0.027) (Figure 2), and for C with T (r = −0.524; *p* = 0.080).

At time T2, moderate correlations were observed between the T:C ratio and the following: the ABK jump (r = 0.405; *p* = 0.191), the DJ jump (r = 0.341; *p* = 0.278), the translation test (r = −0.419; *p* = 0.191), the ability to repeat sprints (r = 0. 389; *p* = 0.211), the agility test (r = −0.386; *p* = 0.215) and the T:C ratio (r = 0.413; *p* = 0.181), and the intermittent endurance test (r = 0.381; *p* = 0.222) and the T:C ratio (r = −0.319; *p* = 0.312). The link between C and T (r = 0.395; *p* = 0.203) also demonstrates this pattern. Simultaneously, it described a strong association (r = 0.543; *p* = 0.068 magnitud) between the amount of time spent on sprint repeatability and C.

Being that of the low correlation ratio, further findings are not displayed in this section (r < 0.3).

## 4. Discussion

The main objective of this study was to examine how total plasma T and C levels and their relationship to their metabolic indicator T:C ratio changed in elite female basketball players over the course of a standard competitive season, in order to observe the gender-specific sport adaptation of the participants with respect to these variables. By identifying the differences in these fluctuations, training and work dynamics were adjusted to the needs of women’s basketball. The hypothesis was that an increase in T levels and a decrease in C levels would cause the outcome of sports performance tests to show significant results.

However, despite this, and as mentioned above, now a days, only a small number of publications have addressed hormonal responses in women’s basketball [40].

The most outstanding results of the present investigation were the observations of the following: (1) A decrease (−7.634%) in C levels; (2) An increase in T levels (1.687%); (3) An improvement in the T:C ratio (26.366%); (4) Significant improvements in jumping (SJ: 11.5%; CMJ: 10.5%; DJ: 13.0); (5) Significant improvement in the results of the intermittent endurance test (Yy (IET): 63.5%) and therefore the relationship established between this test and cardiorespiratory capacity (VO_2max_)_._

Measuring the amount of work performed provides only part of the data on the adaptation of female basketball players [67], and a more complete understanding of players’ adjustments to the imposed load can be achieved by combining various workload monitoring configurations with additional monitoring tools, such as blood levels of T and C hormones [68]. Overall, monitoring these variables can help us better understand how players train and recover [69], which is in line with scientific advice emphasizing how crucial it is to manage external training load to reduce the risk of injury and improve athletes’ physical performance [15].

Studies examining the relationship between hormones and exercise have shown that there is an increase in hormone concentrations in the blood during exercise [70], as the type, intensity, duration of exercise and training level of the individual influence hormone secretion, as it has been shown that skeletal muscle performance can be adversely affected by metabolic problems related to energy production and utilization [27].

One of the most studied hormones in sport is T, a lipid-steroid hormone known for its potent anabolic effect on muscle tissue. As with other hormones, the T concentration increases in response to exercise at a specific intensity threshold, but when exercise is prolonged to exhaustion, it decreases by up to 40% in T [9]. According to recent research, lower T concentrations are linked to increased perceptions of weariness [9], as this hormone is involved in the rate of body regeneration [26]. Research dedicated to studying hormonal variations in female athletes includes research on the influence of exercise on menstrual dysfunction and studies examining the potential impact of menstruation on injury risk. In the case at hand, there has been a slight increase (1.7%) in the levels of T. In our study, although the *p*-value suggests that this change is not significant, the observed increase indicates that the efforts made between these times have been adequate, albeit limited, as in other studies already carried out. With the improved presence of T in the participants’ bodies, it is likely that they will more readily develop muscle mass and improve their athletic performance [71]. Muscle growth and strength will rise in direct proportion to T secretion (within defined theoretical limits) and physical activity level. As a result, given that muscles consume most of the energy, participants will experience up to a 30% increase in energy expenditure during physical activity [72]. In addition, this hormone will increase muscle growth and help reduce fat by accelerating metabolism [72]. Significant and even high percentage changes have been observed in strength and power tests such as SJ, CMJ, DJ and MBT, in addition to a significant change in cardiorespiratory fitness improvement as a major factor transferred from the results obtained from the intermittent endurance test (Table 3). Furthermore, in addition to being statistically significant, these improvements have practical significance, as athletes trained in jumping-based sports, such as basketball, place great importance on this skill.

This catabolic hormone, measured in blood levels, breaks down proteins and lipids before mobilising energy substrates during physical activity [1]. This mechanism is essential for metabolic processes during exercise and for the athlete’s full recovery [1]. However, although high C levels are usually associated with increased protein degradation (catabolism), several studies show that during exercise, muscle activity inhibits the catabolic action of glucocorticoids [73].

Apart from the metabolic effects linked to volume, intensity, and rest interval, it has been demonstrated that training status directly affects the level of the adrenal cortical response, affecting both physical and psychological stress [74]. This mechanism is essential for metabolic processes during exercise and for the athlete’s full recovery [19]. However, even though high cortisol levels are often associated with increased protein degradation (catabolism), several studies have shown that during exercise, muscle activity inhibits the catabolic action of glucocorticoids.

Due to the catabolic role of C, a decrease in C, although not significant, over the course of a training programme, as is the situation here (−7.6%), might indicate a reduced rate of tissue deterioration and help the anabolic environment get better overall with exercise [75]. Thus, a rise in anabolic androgenic activity brought on by training may be shown by the increase in T and the reduction in C. The fact that we discovered that a substantially high correlation is one element to which we can connect (r = 0.633; *p* = 0.03) with the DJ jump at T1 and with the RSA test at T2, although at this time, it is not significant.

Because of this, the decline in C levels might indicate that the kind of basketball training causes adaptation in skilled female athletes and guards against overtraining and overexertion [76]. As stress on the body is likely to be cumulative [18], this may result in mental exhaustion, which may impair one’s ability to function cognitively (decision making) as well as technically (free throws, three-point shooting) [24], especially as competition schedules sometimes increasingly include events several times a week [16,17]. Both factors are essential to identifying the best solutions and minimising any potential drawbacks to competition and training. We can lessen any negative effects by modifying training to guarantee sufficient recovery processes, preventing tiredness accumulation, or making up for lack of stimulus [22].

It can, therefore, be concluded that this C profile, indicative of a decrease in catabolic processes, can be interpreted as an adaptive response to training. When aerobic exercise is performed at the appropriate intensity and duration, cortisol levels will increase. Cortisol levels, however, will drop if the activity intensity is low enough to allow metabolic clearance to surpass adrenal secretion [2]. Given the circumstances, this drop (−7.6%) would indicate a reduced degree of tissue degradation and help the anabolic environment improve overall with exercise [77].

Therefore, T and C are two hormones commonly used as indicators of anabolic and catabolic metabolism, as well as potential markers of physiological stress associated with training [78]. However, the relationship between these hormones and performance or fatigue is still unclear and should be approached with caution [79]. During stressful times, the anabolic-catabolic balance has been monitored using the T:C ratio [25], and this parameter usually varies in relation to fatigue [9] and has been proposed to regulate the intensity of training [74]. As previously mentioned, overtraining may be indicated by a T:C ratio reduction of more than 30% from baseline values since there is currently evidence that a major factor regulating technical performance is the tiredness state [14]. Hypothetically, a decrease in the ratio would indicate a dominance of catabolic processes, which could lead to reduced performance and possible health implications, while an increase would indicate a dominance of anabolic processes [80]. At most, this metric can be regarded as an indirect indicator of the anabolic-catabolic condition of skeletal muscle, notwithstanding the possibility that this interpretation is overly straightforward.

Usually, a rise in C is responsible for the ratio’s decline [81]. There have been no discernible changes in the T:C ratio (T and C in nMol/L) in basketball research. The significant variability of the training pattern—in which duration, volume, and intensity are crucial—strictly influences the T/C ratio, which may not be a valid predictor of overtraining in endurance exercise [28]. In the present case, although the change in this relationship is not significant, the percentage change is large (26%), and a high correlation with performance tests has not been found. Numerous studies found percentage increases that matched those in the current study, and they came to the conclusion that tracking the parameters (T and C) would be extremely helpful in preventing the stress that comes with a basketball season and in better managing the recuperation times [82], as recommended to improve training prescription and even assess the training potential of female athletes [11,12] since it has been observed that repeated exercise without an adequate recovery period can lead to a lasting deterioration of this ratio [30].

Researchers, coaches, and athletes should, therefore, carefully regulate the internal training load in order to maximise athletic performance, minimise unfavourable outcomes, and eventually keep players from overtraining [20]. Since no discernible changes in any of the hormones were found, it would be essential to analyse the hormonal and psychological factors together using an integrative method [83].

## 5. Conclusions

We have different conclusions about this hormone test, especially with female basketball players. Women have more complicated hormonal systems that can be influenced by things like their menstrual cycle, stress, and how hard they train. Stress from a tough season may be a big reason for these inconsistent results. Fatigue can mess up the hormones of female athletes.

Therefore, the regular monitoring of testosterone and cortisol levels would be useful for coaches and trainers. This would allow them to prevent episodes of excessive stress and to properly control players’ recovery periods, thus optimising their performance and avoiding overload-related injuries.

## 6. Limitations

The application of this study is limited due to the controlled environment and profile of the participants. None of the subjects were sedentary, and all were engaged in regular physical activity, so it is unclear whether the findings relate to high levels of exercise in general or specifically to high-level sports performance. In addition, the lack of data on factors such as sleep, and diet precludes a full explanation of the varying levels observed.

## 7. Future Research Lines

It is suggested that future studies contrast these differences in hormone levels with tangible indicators of training load and/or other factors that affect hormonal systems to assess their relationship to athletic performance. However, further research is needed to understand more clearly the hormonal effects of sporting practices.

## 8. Practical Applications

From the results obtained, it was determined that monitoring hormonal responses during seasonal planning can provide valuable information on the level of stress caused by sports training and competition, as well as how athletes adapt to the stress generated by sporting activities. This information can be used to more efficiently manage the training load and plan training cycles throughout the competition season.

## Figures and Tables

**Figure 1 jfmk-09-00133-f001:**
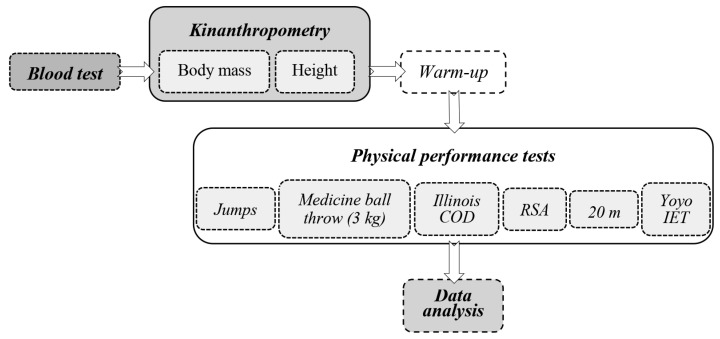
Order of testing (5 min pause between attempts).

**Figure 2 jfmk-09-00133-f002:**
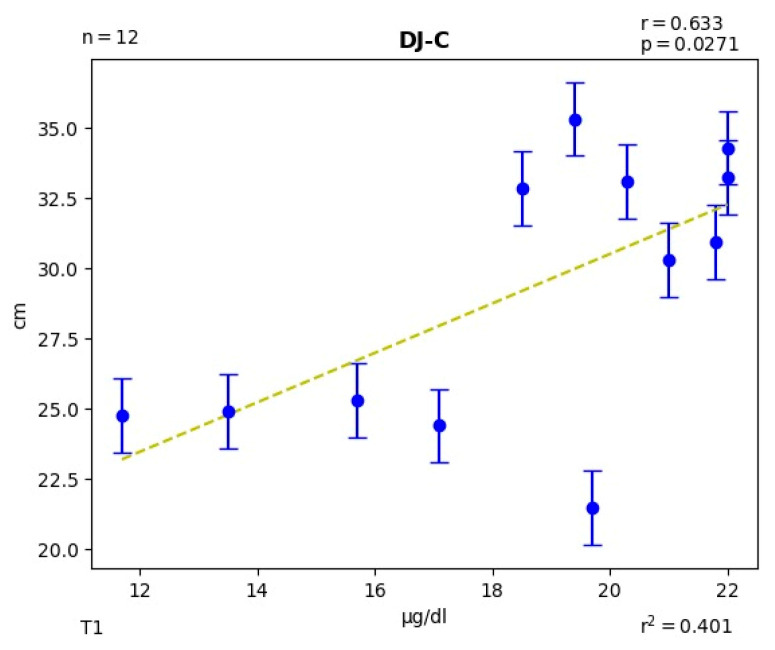
Significant correlation between DJ (cm) and C (μg/dL) levels in T1. The dotted line is the trend line in the relationship between jumping ability and blood Cortisol levels and each of the blue marks represents each of the participants.

**Table 1 jfmk-09-00133-t001:** General data of the participants.

**Age** (yr)	26.0 ± 5.9
**Body mass** (kg)	77.808 ± 12.4
**Height** (cm)	177.6 ± 6.4
**Sports Experience** (yr)	14.7 ± 2.9
**Experience at Elite level** (yr)	5.0 ± 1.1

**Table 2 jfmk-09-00133-t002:** Hormonal Changes in percentage and absolute terms.

	Cortisol (μg/dL)	Testosterone (ng/mL)	T:C Ratio
**T1 (*n* = 12)**	18.6 ± 3.4	0.69 ± 0.2	4.0 ± 2.4
**T2 (*n* = 12)**	17.1 ± 5.0	0.70 ± 0.1	5.1 ± 4.3
** *t* **	2.201		
**%Δ**	−7.634	1.687	26.366
** *p* **	0.326	0.779	0.435
** *η^2^_p_* **	0.331	−0.061	−0.308
**Change Magnitude**	Moderate	Minimal	Moderate
**Probabilistic Inference**	Probably beneficial	Probably trivial	Probably beneficial

T1: Time 1; T2: Time 2; *t*: *t* de Student; the %Δ was calculated as [(T2 − T1)/T1] × 100; *p*: *p*-value; *η*^2^*_p_*_:_ effect size.

**Table 3 jfmk-09-00133-t003:** Physical fitness characteristics in two different moments.

	SJ (cm)	CMJ (cm)	ABK (cm)	DJ (cm)	MBT (m)	20m (s)	RSA (s)	Illinois (s)	Yo-Yo (m)	VO_2max_ (ml·kg·min)
**T1 (*n* = 12)**	29.3 ± 4.5	29.4 ± 5.4	34.3 ± 5.0	29.2 ± 4.8	7.3 ± 0.9	3.5 ± 0.2	7.9 ± 0.4	18.5 ± 0.8	401.7 ± 387.7	50.8 ± 5.3
**T2 (*n* = 12)**	32.7 ±5.2	32.5 ± 4.3	33.6 ± 3.9	33.0 ± 5.2	7.7 ± 0.9	3.4 ± 0.2	7.8 ± 0.4	18.8 ± 0.6	656.7 ± 247.4	54.2 ± 3.4
** *t* **	2.201									
**%Δ**	11.5 ± 16.6	10.5 ± 16.5	−2.1 ± 7.9	13.0 ± 16.6	5.4 ± 8.2	−1.7 ± 5.9	−2.2 ± 3.7	1.6 ± 4.3	63.5 ± 115.6	6.8 ± 2.5
** *p* **	0.029	0.030	0.388	0.010	0.030	0.303	0.066	0.228	0.010	0.008
** *η* ^2^ * _p_ * **	−0.693	−0.633	0.160	−0.764	−0.431	0.265	0.450	−0.426	−0.784	−0.784
**Change** **Magnitude**	Strong	Moderate	Minimal	Strong	Moderate	Moderate	Moderate	Moderate	Strong	Strong
**Probabilistic** **Inference**	Almost certainly beneficial	Almost certainly beneficial	Possibly harmful	Almost certainly beneficial	Possibly beneficial	Possibly beneficial	Possibly beneficial	Possibly harmful	Almost certainly beneficial	Almost certainly beneficial

T1: Time 1; T2: Time 2; *t*: *t* de Student; the %Δ was calculated as [(T2 − T1)/T1] × 100; *p*: *p*-value; *η*^2^*_p_*_:_ effect size.

## Data Availability

To guarantee the openness of the article’s findings, the authors have created a data access declaration that makes the data available to the public upon request from the relevant author.

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
