# Peer review of "Interactions between Stress Levels and Hormonal Responses Related to Sports Performance in Pro Women’s Basketball Team"

_jfmk, 2024, doi:10.3390/jfmk9030133_

Round 1

Reviewer 1 Report

Comments and Suggestions for Authors

First of all, I thank you for the invitation.

We thank the authors of the study for the valuable research exercise they have carried out.

The suggestions have been made in the form of comments in the PDF document. It is respectfully suggested to review them, all of them are on the way to improving the document.

Thoroughly review the results, study conclusions and references. These sections need deep corrections.

The summary, introduction, material and method and discussion section require minor corrections.

Authors are invited to strengthen the document. Once again I congratulate you for the excellent work done.

Comments on the Quality of English Language

It is well written

Author Response

Dear Reviewer,

We would like to express our gratitude for your valuable feedback and thoughtful review of our study. Your comments have greatly contributed to the improvement of our research manuscript.

Additionally, we have carefully reviewed the manuscript to ensure that all formatting requirements and referencing guidelines have been met accurately.

We have carefully considered your suggestions and have made the necessary revisions to address the concerns raised. Specifically, we have provided a more comprehensive discussion on the limitations of our study, acknowledging the small sample size and its impact on generalizability. Additionally, we have incorporated your recommendation to formulate more assertive conclusions, dividing them into cognitive considerations and potential applications.

We believe that these changes have strengthened the clarity and impact of our paper. We appreciate your expertise and guidance throughout this process, as it has undoubtedly enhanced the quality of our research.

Once again, thank you for your time and valuable input. We hope that the revised version of our manuscript now meets the standards of the journal and addresses your concerns satisfactorily.

Sincerely,

Álvaro Miguel-Ortega

Reviewer 2 Report

Comments and Suggestions for Authors

The introduction provides a good background of the study design for the effects of testosterone-cortisol balance on sports performance. And present the relevant previous studies for the current study. However, to improve the manuscript the authors need to articulate the limitations of the previous studies.

In the Method, the authors need to do a power analysis for the sample size. I agree that the quality of the participants is good, but for the quantitative analysis, the power analysis is necessary for the study's small sample size.

Lines 168-170 - the authors need to provide rationales for the chosen time. References need to be provided if possible.

Lines 176-177 – the authors need to provide any considerations for the differences.

The authors need to provide rationales for the chosen physical performance tests in the method section or in the introduction section.

In the discussion, the authors need to emphasize how the findings of this study are different from the previous studies or what this study academically adds to the existing literature.

Also, the authors need to add more practical implications of the current study.

Generally, the authors need to check that all references are properly formatted. Plus please double-check typographical errors or inconsistencies in the text.

Author Response

Dear Reviewer,

We would like to express our gratitude for your valuable feedback and thoughtful review of our study. Your comments have greatly contributed to the improvement of our research manuscript.

Additionally, we have carefully reviewed the manuscript to ensure that all formatting requirements and referencing guidelines have been met accurately.

We have carefully considered your suggestions and have made the necessary revisions to address the concerns raised. Specifically, we have provided a more comprehensive discussion on the limitations of our study, acknowledging the small sample size and its impact on generalizability. Additionally, we have incorporated your recommendation to formulate more assertive conclusions, dividing them into cognitive considerations and potential applications.

We believe that these changes have strengthened the clarity and impact of our paper. We appreciate your expertise and guidance throughout this process, as it has undoubtedly enhanced the quality of our research.

Once again, thank you for your time and valuable input. We hope that the revised version of our manuscript now meets the standards of the journal and addresses your concerns satisfactorily.

Sincerely,

Álvaro Miguel-Ortega

In the Method, the authors need to do a power analysis for the sample size. I agree that the quality of the participants is good, but for the quantitative analysis, the power analysis is necessary for the study's small sample size.

Authors: Line 352 (blue in the text) shows the statistical analysis of power and effect size. We hope this answers your request.

Lines 168-170 - the authors need to provide rationales for the chosen time. References need to be provided if possible.

Authors: We have proceeded to explain and reference what has been requested more exhaustively (line 182, in blue in the text).

Lines 176-177 – the authors need to provide any considerations for the differences.

Authors: We have proceeded to explain the request more fully (line 193, in blue in the text).

The authors need to provide rationales for the chosen physical performance tests in the method section or in the introduction section.

Authors: we have added the justification of the physical performance tests chosen in the methods section (line 229, in blue in the text).

In the discussion, the authors need to emphasize how the findings of this study are different from the previous studies or what this study academically adds to the existing literature.

Authors: we have added what we believe this study contributes more specifically to the discussion (line 435, in blue in the text), although we believe we have done so throughout the text. Please let us know if you do not think so.

Also, the authors need to add more practical implications of the current study.

Authors: we have added practical applications that could be used from the data and conclusions presented here (line 581, in blue in the text).

Generally, the authors need to check that all references are properly formatted. Plus please double-check typographical errors or inconsistencies in the text.

Authors: We have revised all the references, updating some of them to the format established by the American Chemical Society, which is the style followed by the journal. If you think there are still errors, please let us know.

Round 2

Reviewer 1 Report

Comments and Suggestions for Authors

Dear authors

I thank the authors for their valuable work in improving the document.

Each of the observations made has been corrected and its publication is suggested.

Good work and I hope that this contribution to knowledge transcends.

Cordially,

Boryi Becerra

Reviewer 2 Report

Comments and Suggestions for Authors

The authors have successfully addressed the comments, and the manuscript has been improved sufficiently to be published. Thanks for the authors' efforts on this manuscript.